# Efficient Adsorption and Catalytic Reduction of Phenol Red Dye by Glutaraldehyde Cross-Linked Chitosan and Its Ag-Loaded Catalysts: Materials Synthesis, Characterization and Application

Chiara Concetta Siciliano [1,†], Van Minh Dinh [2,†], Paolo Canu [1], Jyri-Pekka Mikkola [2,3] and Santosh Govind Khokarale [2,*]

1    Department of Industrial Engineering, University of Padova, Via F. Marzolo, 9, 35131 Padova, Italy
2    Technical Chemistry, Department of Chemistry, Chemical-Biological Centre, Umeå University, S-90187 Umeå, Sweden
3    Industrial Chemistry & Reaction Engineering, Department of Chemical Engineering, Johan Gadolin Process Chemistry Centre, Åbo Akademi University, FI-20500, Turku, Finland
*    Correspondence: santosh.khokarale@umu.se; Tel.: +46-721262291
†    These authors contributed equally to this work.

**Abstract:** In this study, glutaraldehyde cross-linked chitosan support, as well as the catalysts obtained after loading Ag metal (Ag/Chitosan), were synthesised and applied for adsorption and reduction of phenol red dye in an aqueous solution. The Ag/chitosan catalysts were characterised by X-ray diffraction (XRD), scanning electron microscopy (SEM), Fourier transform infrared spectroscopy (FT-IR) and inductively coupled plasma-optical emission spectrometry (ICP-OES) analysis techniques. The catalytic reduction and adsorption performance of phenol red dye with Ag/chitosan and cross-linked chitosan, respectively, was performed at ambient reaction conditions. The reduction of dye was carried out using sodium borohydride ($NaBH_4$) as the reducing agent, while the progress of adsorption and reduction study was monitored with ultraviolet-visible (UV-vis) spectrophotometry technique. The reduction of the phenol red dye varied with the amount of catalyst, the concentration of $NaBH_4$, Ag metal loading, reaction temperature, phenol red dye concentration and initial $p$H of the dye solution. The dye solution with a nearly-neutral $p$H (6.4) allowed efficient adsorption of the dye, while acidic ($p$H = 4) and alkaline ($p$H = 8, 11, 13.8) solutions showed incomplete or no adsorption of dye. The reusability of the Ag/chitosan catalyst was applied for the complete reduction of the dye, where no significant loss of catalytic activity was observed. Hence, the applicability of cross-linked chitosan and Ag/catalyst was thus proven for both adsorption and reduction of phenol red dye in an aqueous solution and can be applied for industrial wastewater treatment.

**Keywords:** cross-linked chitosan; Ag/chitosan catalyst; phenol red dye; adsorption; catalytic reduction; catalyst reusability

## 1. Introduction

Organic dyes such as azo dyes are commercially important organic moieties which are extensively applied in paints, textile, printing, pharmaceutical industry, dyeing, paper and pulp industries [1]. Besides that, these organic dyes are also used as a $p$H-sensitive indicator in classical to advanced analytical chemistry applications for qualitative as well as quantitative measurements [2]. The consumption of azo dyes has increased tremendously, especially in textile industries, since global demand for clothes and garments has surged exponentially in recent decades. The huge amount of these organic dyes and their derivatives are disposed of in water reservoirs as well as in soil and provide a serious threat to the aquatic environment and can be responsible for irreversible and unwanted

ecological changes [3,4]. These dyes are also highly hazardous to human health and terrestrial animals, being non-biodegradable as well as carcinogenic and mutagenic [4]. To circumvent this problem, researchers worldwide have focused on finding out effective and efficient methods to purify the dyes containing industrial effluents either through chemical as well as physical techniques or combinations thereof. In this regard, physical adsorption, membrane-based filtration, ion exchange and catalyst-based, as well as microbial degradation methods, have been fruitfully applied [4–9]. Adsorption and/or catalytic processing based on photocatalysis, reduction, degradation and oxidation are more efficient, selective and economically viable routes for the purification of water contaminated with various organic dyes since these processes can be carried out under mild reaction conditions as well as using easily available materials [8,10–12].

Supported nanometal catalysts comprising of metals such as silver, palladium, gold and platinum stabilised on various heterogeneous supports such as carbonaceous materials, metal oxides or polymeric supports are widely used for the processing of organic dyes in aqueous mediums [13–16]. The supported nanometal catalysts based on silver metal are considered the preliminary choice of researchers for the processing of organic dyes in water since the precursors required for catalyst synthesis are relatively cheap and widely available compared to other precious metals. Besides that, supported silver metal catalysts still maintain high selectivity and reactivity during the catalytic process in line with other precious-metal-based catalysts. Silver metal supported on various heterogeneous supports such as mesoporous carbon, graphene oxide, mesoporous silica, cotton fabric and metal oxides, etc., is used for the catalytic degradation of organic dyes such as methylene orange, methylene red, phenol red, crystal violet, Congo red, Chicago Sky Blue and malachite green dye etc. [10–16]. Adsorption of the organic dyes from their aqueous solutions over the active surface of the catalyst is a critical step where adsorption followed by reduction or degradation of the dye usually takes place during the catalytic process. Like reduction or degradation approaches, the adsorption of the dyes over the reactive surfaces has also been well-studied for the purification of a water-containing dye embedded in it. It also can be considered a cost-effective technique since metal precursors are not required in the materials synthesis. Besides that, the degradation products which used to remain in the aqueous medium after catalytic degradation or reduction can be avoided. In this regard, besides their valuable contribution to the reduction and degradation process, various types of solid materials, including inorganic oxides, biopolymers or bio-based materials, activated carbon etc., have also been applied as adsorbent materials for the adsorption of organic dyes from their aqueous solutions [4,17–19].

Chitosan is a linear and semi-crystalline polysaccharide consisting of repetitive units of N-acetyl D-glucosamine and D-glucosamine in the polymeric chains. It is a biocompatible and biodegradable polymer obtained from partial deacetylation of the natural biopolymer chitin and has emerged in numerous applications as well as the best replacement for fossil-based synthetic polymers [20,21]. Besides being a bio-renewable polymer, chitosan has the ability to perform as a substrate for precious as well as non-precious metal-loaded heterogeneous catalysts, which were found highly robust and versatile for various synthetic organic transformations [22–24]. Silver metal-loaded chitosan (Ag/chitosan) is also used in various biomedical applications, such as wound dressing, considering its intrinsic antibacterial and antimicrobial characteristics in the resultant material [25–27]. Additionally, chitosan and Ag/chitosan have also been effectively explored in the processing of impure water containing organic dyes and other aromatics through adsorption and catalytic reduction techniques, respectively [28–33].

In this report, glutaraldehyde cross-linked chitosan and its Ag metal loaded analogue, i.e., Ag/chitosan, was studied for both adsorption as well as reduction of phenol red dye in its aqueous solution. For the reduction process of dye, NaBH$_4$ was used as a hydrogen source. Phenol red is a water-soluble organic dye and used as *p*H indicator in various processes such as the development of cell cultures, in-home swimming pool tests, as a diagnostic aid for the determination of renal function and estrogenic properties, etc. It has

been shown that phenol red has a mutagenic effect and causes serious eye damage as well as skin and respiratory irritation. To the best of our knowledge, a detailed study regarding the reduction and adsorption of phenol red dye with chitosan as well as Ag/chitosan catalysts has not been carried out. In this study, initially, the glutaraldehyde cross-linked chitosan was prepared and used as a substrate for the synthesis of Ag/chitosan catalyst, and the obtained materials were characterised by various spectroscopic techniques such as XRD, FTIR and SEM analyses. Further, these chitosan-based materials were used for the reduction and adsorption of phenol red dye in an aqueous solution, whereupon the influence of different reaction parameters on the reaction progress was confirmed. The change in the amount of dye as the reaction proceeded was monitored by a UV-visible spectrophotometer. Finally, the recyclability of the Ag/chitosan catalyst in the reduction of the dye was also described.

## 2. Materials and Methods

### 2.1. Materials

Chitosan with medium molecular weight and 75–85% degree of deacetylation, phenol red (powder, ACS reagent) and glutaraldehyde in aqueous solution (50 wt%) were purchased from Sigma Aldrich (Saint Louis, MO, USA). $NaBH_4$ (powder, 98%) was purchased from Fisher Scientific; silver nitrate, $AgNO_3$ (99%, crystal), methanol (100.0%), ethanol absolute (99.95%), acetone (100.0%), sodium hydroxide and NaOH (98.8%, pellets) were obtained from VWR chemicals. Acetic acid (glacial, 100%) was purchased from Merck. All chemicals were used without further purification.

### 2.2. Instrumentation and Characterisation

The X-ray diffraction (XRD) analysis was obtained on PANalytical X'Pert3 Powder Diffractometer (Malvern Panalytical, Malvern, Worcs, UK) in the 20 angle range of 10–70° with a scan rate of 1°/min using Cu Kα radiation. Fourier transform infrared spectroscopy (FT-IR) analysis of the neat chitosan, cross-linked chitosan with and without adsorbed dye and Ag metal loaded catalyst were carried out with a Bruker Vertex 80 FTIR spectrometer in the range between 500 and 4000 $cm^{-1}$. The morphologies of the Ag metal-loaded catalysts were analysed with scanning electron microscopy (SEM) and energy dispersive X-ray spectroscopy (EDX) technique using a Zeiss Merlin FEG-SEM instrument (Oberkochen, Germany) equipped with an in-lens secondary electron detector. The amount of silver metal on the catalyst was calculated through Agilent 5800 inductively coupled plasma-optical emission spectrometry (ICP-OES, Agilent Scientific Instruments, Headquarters, Santa Clara, CA, USA). The UV spectra to evaluate the progress of reactions were obtained using a UV-3100PC spectrophotometer (VMR International BV, Geldenaaksebaan, Leuven, Belgium).

### 2.3. Preparation of Cross-Linked Chitosan and Ag/Chitosan Catalysts

2.3.1. Cross-Linked Chitosan Support Preparation

The catalyst was prepared following a previously reported method [28]. 0.25 g of chitosan was mixed gently in 12.5 mL of an aqueous solution of acetic acid (2 wt.% $CH_3COOH$), and the mixture was held under stirring for 24 h at room temperature. The amount of added chitosan was dissolved completely in the acidic solution, and a homogenous, transparent and viscous solution was obtained. The chitosan solution was slowly added to the previously prepared alkaline water and methanol solution (distilled water, 17 mL; methanol, 25 mL and NaOH, 5 g) and beads consisting of neutralised chitosan formed and settled at the bottom of the flask. The alkaline solution with beads was kept at room temperature for 24 h. The chitosan beads were separated from the alkaline solution by vacuum filtration and washed with distilled water until neutral *p*H was reached. The beads were further exposed for the cross-linking process, where the beads were added to an alcoholic solution of glutaraldehyde (0.1 mL of 25% of glutaraldehyde solution in water and 6 mL of methanol), and the reaction mixture was refluxed for 6 h. The beads were separated from the alcoholic solution by vacuum filtration and washed with 50 mL of

ethanol and water mixture (50:50 vol.%). The beads were freeze-dried (Scanvac CoolSafe Freeze Dryer) overnight after mixing with 10 mL distilled water, and the obtained material was kept in a desiccator prior to the synthesis of Ag/chitosan catalysts.

### 2.3.2. Preparation of Ag/Chitosan Catalysts

The Ag/catalyst was prepared from $AgNO_3$ as a precursor and previously prepared glutaraldehyde cross-linked chitosan, where the amount of $AgNO_3$ was varied during the synthesis. The 0.2 g of freeze-dried chitosan beads were mixed with 10 mL aqueous solution of $AgNO_3$ containing either 0.04, 0.02 or 0.005 g of Ag precursor, and the mixture was heated at 70 °C overnight with a reflux condenser. The catalyst beads were filtered by vacuum filtration and washed with distilled water ($3 \times 20$ mL), and the recovered solid was further freeze-dried and stored in a desiccator before its characterisation and catalytic applications. The ICP-OES analysis showed that the catalysts were obtained with 1.4, 4.4 and 6.7 wt.% of Ag metal in their composition after the use of 0.005, 0.02 or 0.04 g of $AgNO_3$, respectively, during the synthesis. The catalysts will be here onwards designated as 1.4 wt.%Ag/chitosan, 4.4 wt.%Ag/chitosan and 6.7 wt.%Ag/chitosan in upcoming descriptions regarding their characterisation and application in catalytic measurements.

### 2.4. Catalytic Phenol Red Dye Reduction

The catalytic reduction of phenol red dye in its aqueous solution was carried out with Ag/chitosan catalyst in a 5 mL glass vial under various reaction parameters. For the different concentrations of dye, 6 mg of 6.7 wt.%Ag/chitosan catalyst and 5 mg of $NaBH_4$ were mixed with 2 mL of 0.001, 0.003 or 0.005 M solution of phenol red dye and the reaction mixture was stirred at room temperature (20 °C) for the desired reaction time. After the addition of the catalyst and $NaBH_4$, the required 0.1 mL of the reaction mixture was periodically transferred to a quartz cuvette and diluted with 1 mL of distilled water, where the absorbance of the resulting solution was measured on the UV-vis spectrophotometer. The volume of the samples taken from the reaction mixtures varied with the concentration of phenol red dye solution to maintain the absorbance value near unity. The conversion of phenol red dye was calculated by the absorbance value obtained with decreasing peak at 571 nm ($\lambda_{max}$) and Equation (1).

$$\% \text{ Phenol red dye conversion} = \frac{A_0 - A_t}{A_0} \times 100 \qquad (1)$$

where $A_0$ and $A_t$ are the absorbance values at time zero and time t, respectively. The $A_0$ value of the absorbance was measured with the alkaline solution of phenol red dye. In this case, the solution of dye with various concentrations, such as 0.001, 0.003 or 0.005 M, was prepared with 0.1 N NaOH solution. In the case of the different amounts of the catalyst, 1, 2, 4 or 6 mg of 6.7 wt.%Ag/chitosan catalyst with 5 mg of $NaBH_4$ each were exposed to 0.001 M dye solution. For the influence of different amounts of $NaBH_4$, 1, 2.5 or 5 mg and 6 mg of 6.7 wt.%Ag/chitosan catalyst were used with 0.001 M phenol red dye solution. The influence of the reaction temperature on the reduction of phenol red dye was also examined, ranging from room temperature (20 °C) to 30 and 40 °C whereupon a dye solution with 0.001 M concentration, 6 mg Ag/chitosan and 5 mg NaBH4 was applied in the process. For the influence of the initial *p*H of the dye solution, the 6 mg 6.7 wt.%Ag/chitosan catalyst and 5 mg of $NaBH_4$ were added to solutions at initial *p*Hs of 4–11. Initially, an aqueous solution with different *p*Hs, such as 4, 8 and 11, was prepared by mixing 0.1 N HCl and 0.1 N NaOH appropriately and thus, obtained solutions were used to prepare 0.001 M phenol red dye solution. The *p*H levels of the 0.001 M solution of phenol red dye obtained with deionised water and 0.1 N NaOH were observed as 6.4 and 13.8, respectively.

### 2.5. Phenol Red Dye Adsorption

The adsorption of phenol red dye from its aqueous solution with different initial *p*H was carried out with cross-linked chitosan support where 6 mg of the support material was

mixed with 2 mL of 0.001 M solution of dye. In a separate study, after adsorption of the dye from its solution with nearly neutral *p*H, i.e., 6.4, 5 mg of NaBH$_4$ was added to the reaction mixture for further reduction of the adsorbed dye. The progress of the adsorption and reduction of phenol red dye with support material was confirmed by the previously described method and UV-vis spectrophotometry technique.

### 2.6. Recyclability of the Ag/Chitosan Catalyst

The 6.7 wt.%Ag/chitosan catalyst (6 mg) was used to reveal the recyclability of the catalyst for the reduction of phenol red dye. After the complete reduction of the dye, the catalyst was separated from the reaction mixture by vacuum filtration using a nylon filter paper. The catalyst was washed several times (10 mL × 3) with distilled water, and a moist catalyst, along with a new batch of 5 mg of NaBH$_4$, was further mixed with the 2 mL fresh aqueous solution of phenol red dye (0.001 M). The reaction mixture was then stirred for 18 min, and the progress of the reaction and conversion of the dye was carried out according to the methods described in the previous sections. The catalyst was recycled additionally four times more for the phenol red dye reduction following a similar procedure. After the fifth recycle, the recovered catalyst was freeze-dried and the obtained solid was further analysed by ICP-OES analysis to confirm the Ag metal leaching during the recyclability study.

## 3. Results
### 3.1. Characterisation of Cross-Linked Chitosan and Ag/Chitosan Catalysts

The structural properties of the cross-linked chitosan and Ag/chitosan catalysts were analysed by powder XRD analysis and the obtained XRD patterns are shown in Figure 1A.

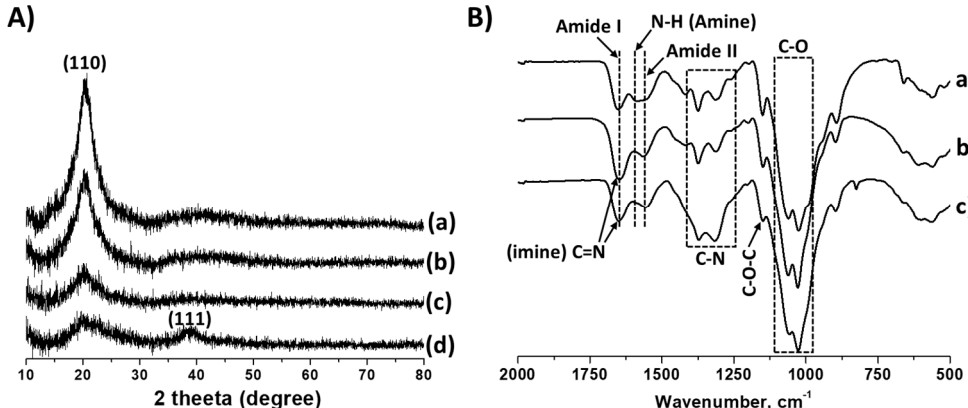

**Figure 1.** (**A**) XRD pattern of cross-linked chitosan (a) cross-linked chitosan, Ag/chitosan catalysts with (b) 1.4, (c) 4.4 and (d) 6.7 wt.% of Ag. (**B**) FT-IR spectrum of (a) neat chitosan, (b) cross-linked chitosan and (c) 6.7 wt.%Ag/chitosan catalyst.

A single characteristic peak assigned to the crystal plane (110) was observed at 2θ value of 21.8° in cross-linked chitosan, whereas the peak at 11.7° was not observed. The reason might be the formation of amorphous solids during the acid-base treatment and glutaraldehyde-induced cross-linking process [27]. For Ag/chitosan catalysts, as the amount of Ag metal increased, the intensity of the peak for the crystalline plane (110) steadily decreased, perhaps due to the covering by metal particles. For the catalyst with 6.7 wt.%Ag metal loading, a broad and low-intensity peak at 2θ value of 21.8° for (110) plane was obtained while a new characteristic peak corresponding to the crystal plane (111) for metallic Ag was observed at 37.9° [27]. However, the peak for the crystalline phase assigned to metallic Ag was not observed in the case of catalysts with metal loadings 1.4 and 4.4 wt.%. Considering the probable homogeneous distribution of metal particles over the support material, the amount of Ag metal may be below the detection limit.

The neat and cross-linked chitosan, as well as Ag/chitosan catalyst with 6.4 wt.% of Ag loading, were also characterised with FT-IR analysis to understand the structural changes in chitosan after cross-linking and Ag metal loading. As shown in Figure 1B, the neat and partially deacetylated chitosan displays the characteristic band at 1585 cm$^{-1}$ for N–H stretching vibrations in the amine group as well as bands at 1655 and 1564 cm$^{-1}$ belonging to C=O (amide I) and N–H (amide II) bending vibration in the amide group, respectively [34]. The bands for the stretching vibration of the C–N bond were observed between 1376 and 1255 cm$^{-1}$ while the band belonging to 1, 4-glycosidic bond (C–O–C) as well as C–O bond in secondary (C$_3$) and primary (C$_6$) –OH groups of chitosan appeared at 1150, 1060 and 1025 cm$^{-1}$, respectively. After cross-linking with glutaraldehyde, the band attributed to N–H bending vibrations in amine disappeared, and the band belonging to stretching vibrations in the C=N bond, which formed between the amine group of chitosan and carbonyl group of glutaraldehyde appeared at 1650 cm$^{-1}$. This newly formed band overlaps with the band of the amide I functional group (C=O in the amide group); hence the overall intensity of the band increased [35]. After Ag metal loading, significant changes in the functional groups were not observed except for the increase in the intensity of the band attributed to C-N stretching vibrations.

The SEM images and SEM-EDX spectrum of the cross-linked chitosan and the Ag/chitosan catalysts with different Ag metal loadings are shown in Figure 2. The SEM images show that material with a rough surface was obtained after the cross-linking of the chitosan with glutaraldehyde, while materials with identical morphology were obtained after various amounts of silver loading (Figure 2A). However, the SEM-EDX analysis confirms the deposition of the Ag metal over the chitosan support, and the intensity of the peak of metallic Ag increased with increasing metal loading (Figure 2B).

The XRD and SEM-EDX analysis confirmed the formation of Ag nanoparticles on the cross-linked chitosan support without the use of any additional reducing agents. It was previously reported that chitosan is considered a non-toxic, mild reducing as well as stabilising agent in Ag metal nanoparticle synthesis [36]. In addition, chitosan also performs as a capping agent to control the growth of nanoparticles and helps to avoid their agglomeration. After mixing an aqueous solution of Ag$^+$ ion salt with chitosan, the oxygen and/or nitrogen atom of hydroxyl and amine groups in the chitosan chains, respectively, serve as a ligand and coordinate with the metal ions. Under thermal treatment, hydroxyl and amine (or amide groups in cross-linked chitosan) groups in chitosan further reduce the Ag$^+$ ions and stabilise the resultant Ag metal nanoparticles (Figure 3) [37]. In the actual reduction approach, the oxygen or nitrogen atom may lose their electrons and transform to their oxidised form upon reduction of the Ag$^+$ ions to Ag nanoparticles.

**(A)**

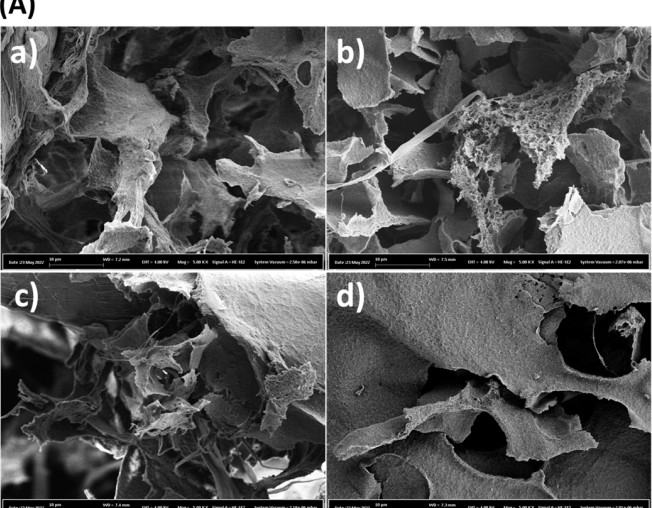

**Figure 2.** *Cont.*

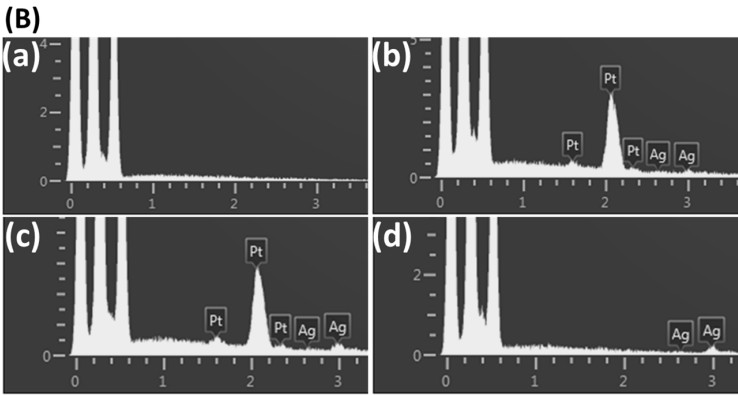

**Figure 2.** (**A**) SEM images, and (**B**) SEM-EDX spectrum of (a) cross-linked chitosan, Ag/chitosan catalysts with (b) 1.4, (c) 4.4 and (d) 6.7 wt.% of Ag.

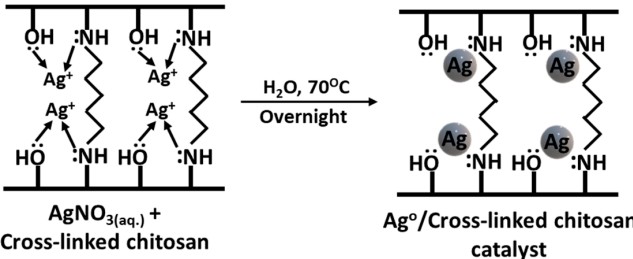

**Figure 3.** Synthesis of Ag/chitosan catalyst from Ag salt and cross-linked chitosan.

### 3.2. Reduction of Phenol Red Dye

The catalytic reduction of phenol red dye was carried out with Ag/chitosan catalysts and NaBH$_4$ as a hydrogen source. In this case, the catalytic reduction of the phenol red dye was monitored with a UV-visible spectrophotometer, where the influence of various reaction parameters on the activity of the catalyst was studied.

### 3.2.1. Influence of Amount of Catalyst

Initially, the influence of the amount of Ag/chitosan catalyst on the reduction ability of the catalyst was studied where prior to the experiment, 6 or 2 or 1 mg of 6.7 wt.%Ag/chitosan catalyst and 5 mg of NaBH$_4$ were added in 2 mL of 0.001 M aqueous solution of phenol red dye. It was observed that after the addition of Ag/chitosan catalyst and NaBH$_4$ in the red-orange coloured phenol red dye solution, the colour of the reaction mixture turned to fuchsia (colour in between pink and violet) (Figure 4A,B). As the reaction progressed, the fuchsia colour steadily disappeared with time, and a colourless solution was obtained (Figure 4C).

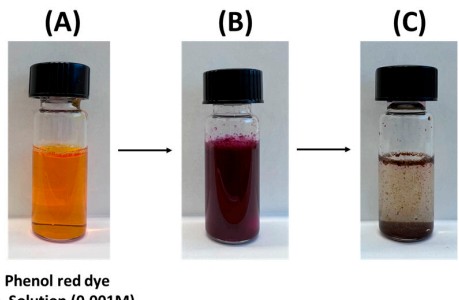

**Figure 4.** (**A**) phenol red dye aqueous solution (0.001 M), the reaction mixture (**B**) after the addition of 6 mg of 6.7 wt.%Ag/chitosan catalyst and 5 mg of NaBH$_4$ and (**C**) after complete reduction of phenol red dye.

As shown in Figure 5A, the 0.001 M red-coloured aqueous solution of phenol red dye gives rise to a peak with a $\lambda_{max}$ value of 436 nm. However, after the addition of the catalyst and $NaBH_4$ and followed by a reaction time of 3 min, the peak at 436 nm disappeared, whereas a peak with a comparatively lower intensity appeared at $\lambda_{max}$ value of 571 nm. The aqueous solution of phenol red dye is usually $p$H sensitive in terms of colour and is frequently used as an indicator in commercial cell culture media. The aqueous solution phenol red dye shows yellow colour in the acidic $p$H range, i.e., below $p$H 6.5, and red-orange colour in between $p$H 6.5 and 7.5, while it turns to fuchsia in alkaline $p$H, i.e., above $p$H 7.5 [38]. Figure 6 shows the chemical structures of the phenol red dye in its aqueous solution with different $p$H scales, where the dye shows tautomeric and zwitterion form in the high and moderately acidic $p$H ranges of the solution, respectively. At a neutral $p$H, the phenol red dye transforms to its phenol form, whereas at high $p$H, the phenate form of the dye remains dominant [38]. In other words, the phenol red dye forms variable molecular structures with different charge distributions depending on the $p$H of the solution where the –C=C– and –C=O chromophores in these structures impart different colours to the solution through an extended conjugation. In the case of a catalytic process, since the colour of the reaction mixture changed to pale fuchsia after the addition of catalysts and $NaBH_4$, the $p$H of the reaction mixture was over 7.5. The decomposition of $NaBH_4$ in water usually increases the $p$H of the solution, accompanied by the release of $H_2$ gas (reaction Equations (2) and (3)).

$$H_2O \rightleftharpoons H^+ + OH^- \tag{2}$$

$$BH_4^- + H^+ + 3H_2O \rightleftharpoons H_3BO_3 + 3H_2 \tag{3}$$

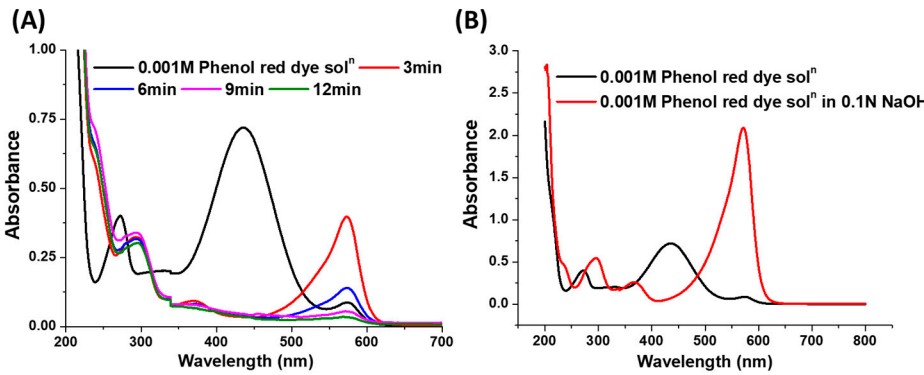

**Figure 5.** (**A**) UV-vis spectra for the reduction of phenol red dye by 6.7 wt.%Ag/chitosan catalyst and $NaBH_4$ (**B**) UV-visible spectra of 0.001M solution of phenol red dye in distilled water and 0.1 N NaOH solution.

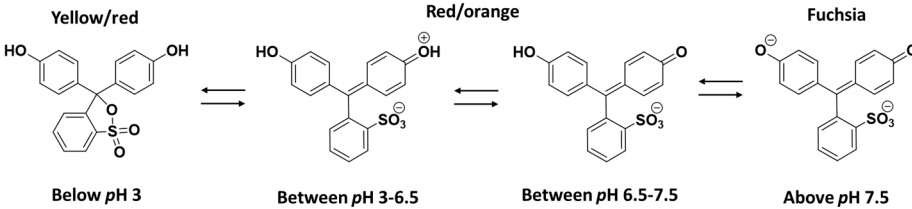

**Figure 6.** Chemical structures of phenol dye in its aqueous solution with different $p$H conditions.

To confirm this, i.e., a change in colour of the reaction mixture with $p$H, a phenol red dye solution in 0.1 N NaOH solution was prepared, and it was observed that the resulting solution became intense fuchsia in colour and a peak with $\lambda_{max}$ value 571 nm was obtained in UV-vis spectroscopic measurements (Figure 5B). Hence, it was shown that after the addition of catalysts and $NaBH_4$, the tautomeric and zwitterion forms of the phenol red dye were converted to their phenate analogue in an alkaline medium. As the

reaction progressed, after the addition of the catalyst and NaBH$_4$, steady decolouration of the reaction mixture occurred due to the reduction of the dye with the Ag/chitosan catalyst and released H$_2$ and, simultaneously, the intensity of the peak at 571 nm also decreased.

Figure 7A shows that the reduction ability of the catalyst steadily decreased with a decrease in its amount. Further, it was observed that NaBH$_4$ is also able to reduce the phenol red dye in the absence of a catalyst, but the rate of the reaction was comparatively low compared to catalyst enhanced process. Since the lower amount of catalyst, i.e., 1 mg, also gave rise to the complete reduction of the dye; the prepared Ag/chitosan catalysts are highly efficient in the reduction of phenol red dye.

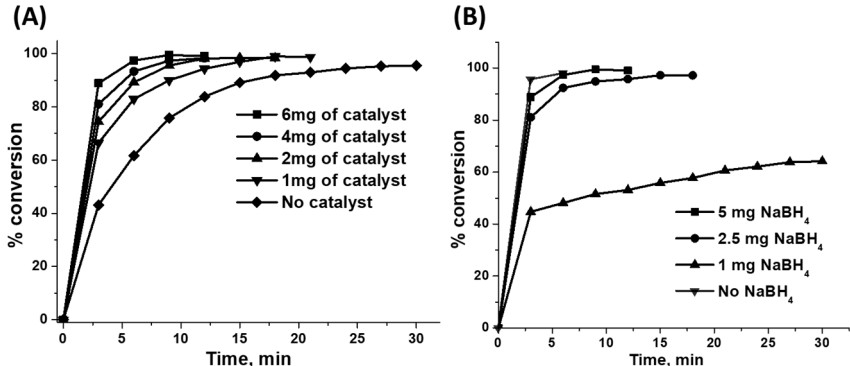

**Figure 7.** Reduction of phenol red dye with 6.7 wt.%Ag/chitosan catalyst and NaBH$_4$ (**A**) different amounts of catalyst and 5 mg of NaBH$_4$ and (**B**) different amounts of NaBH$_4$ and 6 mg of 6.7 wt.%Ag/chitosan catalyst.

### 3.2.2. Influence of Amount of Reducing Agent, NaBH$_4$

Since NaBH$_4$ was used as the reducing agent in the reduction process of phenol red dye, its amount could influence the progress of the reaction in terms of the amount of H$_2$ released. As shown in Figure 7B, the rate of the reduction of dye was significantly enhanced with an increase in the amount of NaBH$_4$ from 1 mg to 2.5 and 5 mg in the reaction mixture. With 1 mg of the reducing agent, no complete reduction of dye was observed, whereupon merely 60% of the phenol red dye was reduced even though the reaction proceeded for 30 min. However, a nearly complete reduction of dye was observed after the use of 2.5 and 5 mg NaBH$_4$, within 18 and 9 min, respectively. Surprisingly, without the use of NaBH$_4$, the rate of discolouration of phenol red dye was found to be high compared to reaction mixtures where a reducing agent was applied, and a colourless reaction mixture was obtained within 6 min (Figure 8A). As depicted in Figure 8B, a catalyst with violet colour was obtained when it was separated from the reaction mixture by filtration. After the addition of NaBH$_4$, the dye desorbed from the catalyst, the supernatant solution in the reaction mixture became pale fuchsia, and it transformed into a colourless liquid in 9 min of reaction time. A more detailed description of the adsorption of the phenol red dye is mentioned in the upcoming sections.

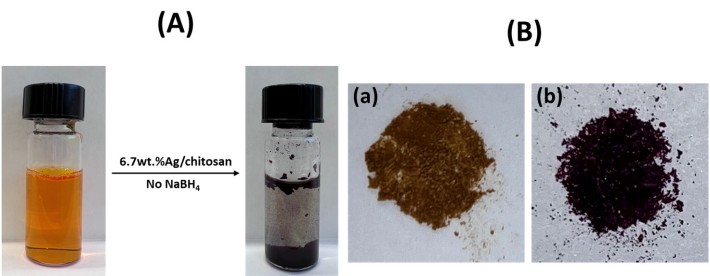

**Figure 8.** (**A**) Addition of 6.7 wt.%Ag/chitosan catalyst in phenol red dye solution, and (**B**) (a) fresh Ag/catalyst and (b) catalyst with adsorbed phenol red dye.

### 3.2.3. Influence of Concentration of Phenol Red Dye Solution

The catalytic activity was further examined upon the reduction of the phenol red dye, whereupon dye solutions with different concentrations, such as 0.001, 0.003 and 0.005 M, were used. The dye reduction ability of the catalyst was not considerably varying with different concentrations containing dye solutions when 5 mg of $NaBH_4$ was applied (Figure 9A). However, after reducing the amount of $NaBH_4$ to 2.5 mg, the rate of the reduction varied with different concentrations of the phenol red dye solution. As shown in Figure 9B, the dye conversion rate steadily decreased as its initial concentration in the aqueous solution increased from 0.001 M to 0.005 M. It indicates that the amount of evolved $H_2$ was not sufficient for the reduction of the increased concentration of dye—if 2.5 mg of reducing agent was used.

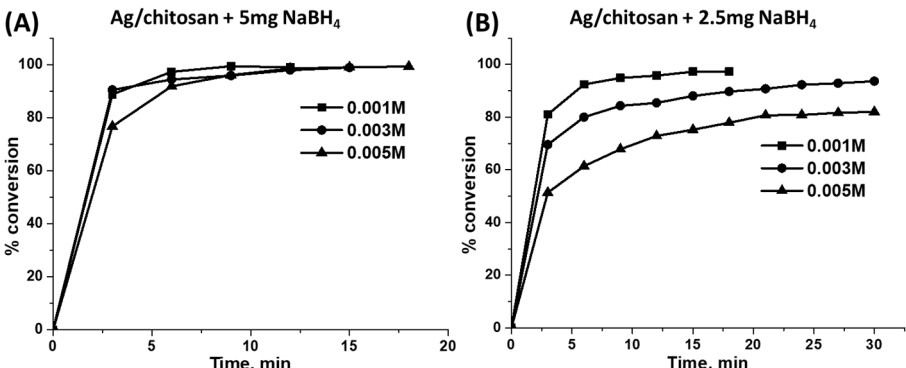

**Figure 9.** Reduction of the phenol red dye at different concentrations in aqueous solutions by 6 mg of 6.7 wt.%Ag/chitosan catalyst and (**A**) 5 mg or (**B**) 2.5 mg of $NaBH_4$.

### 3.2.4. Influence of the Amount of Ag Metal Loadings

In order to understand the influence of Ag metal in the reduction of phenol red dye, Ag/chitosan catalysts with different Ag metal loadings, such as 1.4 or 4.4 or 6.7 wt.%, were chosen during the reduction process. As shown in Figure 10A, the rate of reduction reaction steadily increased with the Ag metal loading on the cross-linked chitosan support. These observations are in agreement with the studies such as powder XRD and SEM-EDAX analysis of catalysts with different metal loadings (Figures 1A and 2B). Further, in the case of cross-linked chitosan, the dye reduced at a slow rate compared to its Ag metal-containing analogues, as shown in Figure 7A, and possibly mainly $NaBH_4$ contributed to the reduction process. Hence, like in the previous study regarding the influence of the catalyst amount, it again confirmed that the Ag metal is necessary for the reduction of the phenol red dye since it is not only involved in the activation of $H_2$ obtained after degradation of $NaBH_4$ but also facilitates a transfer of electron to dye molecules for their subsequent reduction.

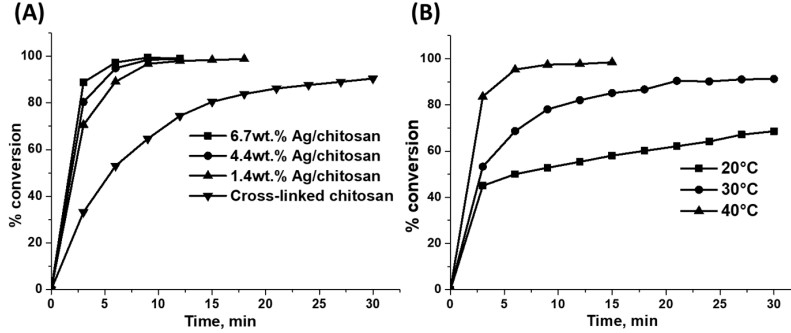

**Figure 10.** (**A**) Reduction of phenol red dye with 6 mg of Ag/chitosan catalysts with different Ag metal loadings or cross-linked chitosan and 5 mg of $NaBH_4$, (**B**) reduction of phenol red dye at different temperatures with 6 mg of 6.7 wt.%Ag/chitosan catalyst and 1 mg of $NaBH_4$.

### 3.2.5. Influence of the Reaction Temperature

Reaction temperatures during the reduction process were also varied to understand their role in the progress of the reaction where the reaction mixture was heated at either room temperature (20 °C) or 30 or 40 °C. Since the reaction rate was high even at room temperature with 5 mg of NaBH$_4$ and 6 mg of 6.7 wt.%Ag/chitosan catalyst, the amount of NaBH$_4$ was reduced to 1 mg in the experiments where the reaction temperature was varied. As shown in Figure 10B, the rate of the reaction increased as the reaction temperature increased from room temperature to 30 and 40 °C.

### 3.2.6. Influence of Initial *p*H of the Phenol Red Dye Solution

Lin et al. previously described that the *p*H of the solution influences the decomposition of NaBH$_4$ and subsequent H$_2$ gas generation during catalytic reduction of *p*-nitrophenol over Au/Fe$_3$O$_4$ catalyst [39]. In this case, the authors explained that the rate of generation of H$_2$ and reduction of *p*-nitrophenol steadily decreased with an increase in the *p*H of the solution, whereas no reduction occurred at *p*H 11. As shown in the reaction Equations (2) and (3), the proton (H$^+$) released after hydrolysis of water reacts with the hydride (H$^-$) anion of NaBH$_4$ and generates H$_2$ gas. The acidic to neutral *p*H condition facilitates the hydrolysis of water resulting in the release of a proton, and then H$_2$ gas generation through the decomposition of NaBH$_4$ becomes feasible. On the other hand, under alkaline conditions, both the rate of hydrolysis of water, as well as decomposition of NaBH4 decreases and diminishes the subsequent rate of H$_2$ generation and reduction of dye. In order to elucidate more about the influence of the initial *p*H of the phenol red dye solution in the reduction process, 6.7 wt.%Ag/chitosan catalyst and NaBH$_4$ was mixed with dye solutions with different initial *p*H such as 4, 6.4, 8, 11 or 13.8. As shown in Figure 11, the reduction ability of the catalysts was not significantly influenced by the initial *p*H of the solution until it reached 11. For 3 min of reaction time, the reduction rate decreased somewhat as the *p*H of the solution increased from 4 to 11, while no significant difference was observed as the reaction proceeded further. Unlike the previously described reduction of *p*-nitrophenol, in this study, the solution with *p*H 11 also allowed an efficient reduction of the dye [39]. Similarly, Wang et al. showed that Ag nanoparticle-entrapped hydrogel prepared from chitosan also enabled a complete reduction of methylene blue and Congo red until *p*H 11 [40]. Furthermore, the reduction of phenol red dye proceeded at a slow rate when a highly alkaline dye solution with *p*H 13.8 was applied.

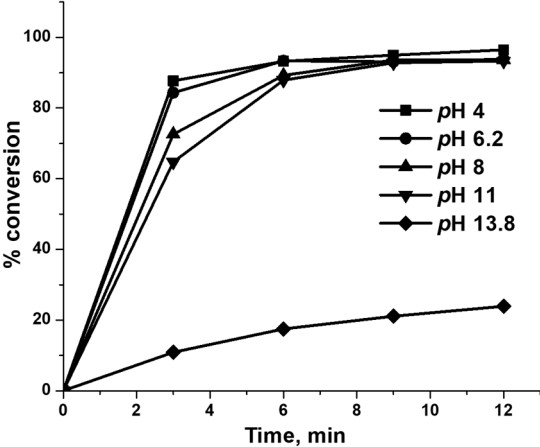

**Figure 11.** Reduction of the phenol red dye in an aqueous solution (0.001 M) with different initial *p*H in the range of 4–13.8. The amounts of 6.7 wt.%Ag/chitosan catalyst and NaBH$_4$ are 6 and 5 mg, respectively.

### 3.3. Adsorption of Phenol Red Dye with Cross-Linked Chitosan Support

3.3.1. Influence of *p*H on the Adsorption of Phenol Red Dye

The *p*H of the dye solution significantly influences its adsorption on the adsorbent surfaces since a change in *p*H alters the charges over the adsorbent and also changes the chemical structure of the dye. Kwok et al. reported that the chitosan surface is sensitive to the *p*H of the solution where adsorption of the arsenate ions effectively occurred in a solution within the acidic *p*H range (below *p*H 7). However, as the *p*H value increased above its neutral value, the rate of adsorption steadily decreased, and the rate of desorption of the adsorbed species increased [41]. The author also described the adsorption-desorption phenomenon of the arsenate ions in terms of the point of zero charge (*p*H$_{pzc}$, the *p*H value at which the surface charge is zero) and surface charge density of the chitosan surface measured by the potentiometric titration method. The *p*H$_{pzc}$ values of the studied chitosan particles were observed to be around 8 since, above this value, the surface becomes negatively charged. Besides that, the surface charged densities measurements also represent that the isoelectric point value was observed at 6.38, whereas the surface charge of the chitosan surface below this value was found to be positively charged. In other words, in the acidic *p*H range, the chitosan surface remains positively charged as the negative charges decrease due to neutralisation which allows efficient adsorption of the arsenate anion until the neutral *p*H of the solution. Wang et al. also showed equivalent observation in the case of adsorption of fulvic acid over a chitosan surface, where the point of zero charge study shows that the adsorption capacity of the adsorbent was high below *p*H 9 while its capacity substantially decreased over this value due to reverse of the charges of the surface from positive to negative [40]. Wahab et al. studied the photodegradation of the phenol red dye over nanocrystalline titanium oxide (TiO$_2$) particles, where it was observed that since the TiO$_2$ surface remained positively charged under acidic conditions (below *p*H 6.5), the rate of degradation of the dye was increased and vice versa in case of alkaline medium [38]. Further, the authors presumed that from acidic to neutral *p*H conditions, the phenol red dye also remained in the zwitterionic form or as a structure with a negatively charged sulfate group which has an electrostatic attraction towards positively charged surface under a similar *p*H range (Figure 5). Ma et al. also applied chitosan as an adsorbent material for the adsorption of Congo red and methylene blue dye, where the process was studied for a dye solution with *p*H 6.5 [5].

In the current study, the influence of the initial *p*H of the dye solution for its adsorption on the cross-linked chitosan support was studied, where dye solutions with varying *p*H from 4 to 13.8 were applied. As shown in Figure 12A, incomplete adsorption of the dye was observed in a solution with *p*H 4, whereupon the peak at 436 nm decreased from its initial value while peak at 571 nm increased. This suggests that the supernatant solution became alkaline since the peak assigned to negatively charged species of the dye appeared. Below *p*H 6, the surface charge of chitosan became positive due to the protonation of amine groups (–NH$_3^+$). As a negatively charged molecule, the phenol red dye eagerly adsorbs on the chitosan surfaces through its electrostatic interaction with positively charged amine groups [42,43]. However, at high concentrations of H$^+$, the dye molecule desorbs from the chitosan surface, which further increases the final *p*H of the solution [41,44]. Furthermore, high adsorption of the dye was observed at *p*H 6.4, whereby nearly complete adsorption occurred in 18 min (Figure 12B). In this case, the dye adsorbed through both physical (hydrogen bonding and van der Waals interaction) as well as electrostatic interaction with the chitosan surface. In the case of the *p*H 6.4 solution, also the peak at 571 nm appeared less intense, suggesting that the reaction mixture became marginally alkaline after dye adsorption due to continuous protonation and deprotonation of the amine groups in chitosan support [41]. In case of an increase in *p*H of the solution to 8, the adsorption capacity of chitosan support for the dye further decreased, and the solution became more alkaline as the peaks at 436 and 571 nm were highly intense compared to reaction mixtures with *p*H 4 and 6.4 (Figure 12C). The negligible adsorption of the phenol red dye observed at *p*H 11 and 13.8 was obvious since the peak intensity at 571 nm did not decrease. As

described previously, above neutral *p*H, the surface charge of chitosan becomes negative, and this disfavoured the adsorption of the negatively charged dye molecules [38,40]. Hence, a dye adsorption study on the chitosan surface demonstrated that an aqueous solution of dye comprised of *p*H close to neutral facilitates the adsorption contrary to acidic and alkaline dye solutions.

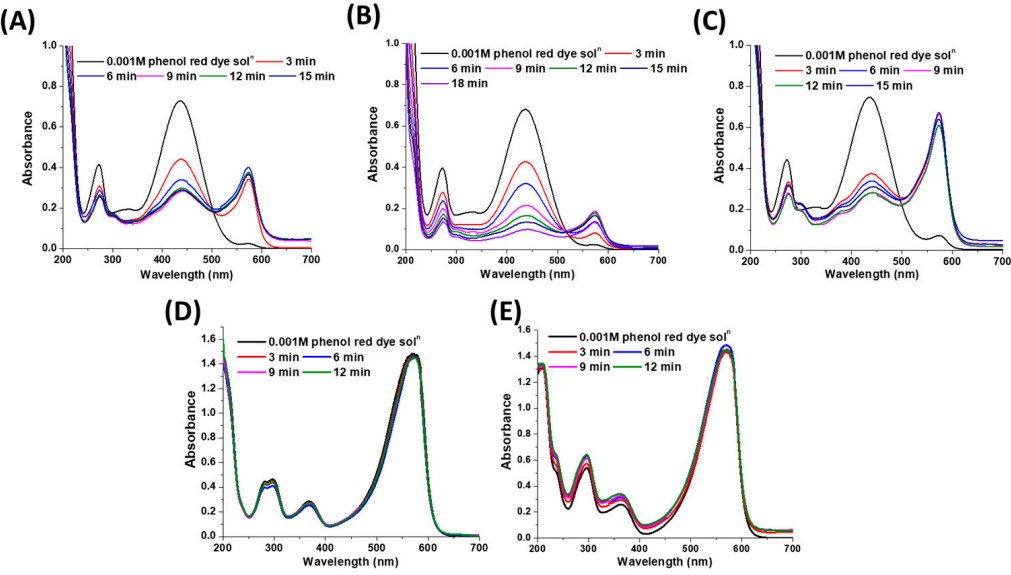

**Figure 12.** Adsorption of phenol red dye by 6 mg of cross-linked chitosan support at various *p*H (**A**) 4, (**B**) 6.4, (**C**) 8, (**D**) 11 and (**E**) 13.8.

### 3.3.2. Adsorption Followed by Reduction of Dye with Cross-Linked Chitosan Support

As shown in Figure 12B, the cross-linked chitosan support gave rise to a high adsorption ability for the phenol red dye over its surface, and the absorbance steadily decreased as the reaction proceeded. The corresponding changes in the absorbance of the phenol red dye are shown in Figure 13A. As shown in Figure 14A, the cross-linked chitosan support displayed yellow colour, which further converted to a violet-coloured solid after the interaction with an orange-red coloured solution of the phenol red dye. Further, the FT-IR analysis of the cross-linked chitosan support with and without the adsorbed dye was also carried out, and the obtained spectra were compared to confirm the dye adsorption over the surface of the support. As shown in Figure 14B, the intensity of the absorption bands regarding various functional groups, including the –OH group (3400 cm$^{-1}$) in cross-linked chitosan support, decreased after the adsorption of dye on the surface, and broadened bands with lower intensity were obtained. Besides that, the characteristic bands attributed to the benzene ring, –OH, carbonyl (–C=O), C–O and sulphonate (SO$_3^-$) groups in phenol red dye molecule also disappeared, and this can be attributed to the homogeneous distribution of dye over the surface of cross-linked chitosan support [38,45].

To examine the reduction of the adsorbed dye, NaBH$_4$ was duly added to the reaction mixture after the adsorption of dye over cross-linked chitosan support. It was observed that the dye desorbed from the chitosan surface, and the reaction mixture became pale fuchsia in colour after the addition of the reducing agent. As shown in Figure 13B, the reduction of the dye proceeded slowly and was mainly due to NaBH$_4$ since it was previously observed that the dye reduced at a slow reaction rate with the reducing agent compared to the Ag/chitosan catalyst involved process (Figure 10A). Here, the adsorption and desorption of dye before and after the addition of the reducing agent, respectively, can be correlated to the change in the *p*H of the reaction mixture since NaBH$_4$ changes the *p*H of the solution to alkaline, which allows the desorption of the dye from the surface of the adsorbent [40,41].

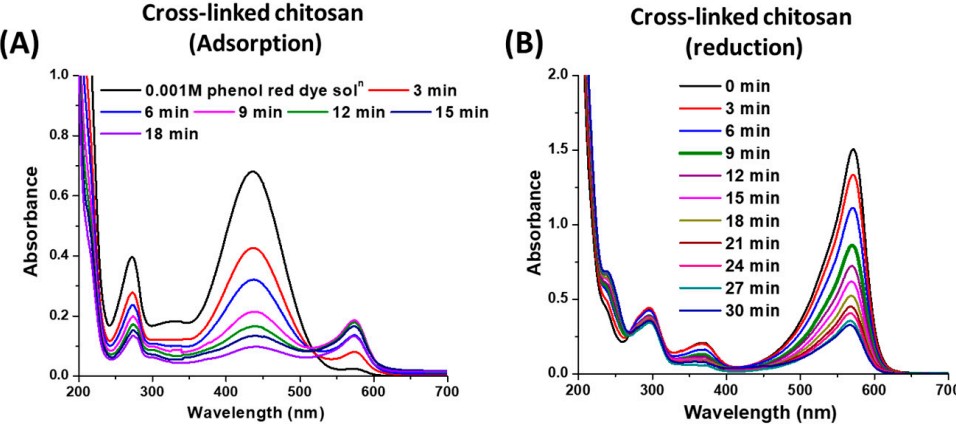

**Figure 13.** (**A**) Adsorption and (**B**) adsorption followed by reduction of phenol red dye with cross-linked chitosan support. The amount of chitosan support and NaBH$_4$ 6 and 5 mg, respectively.

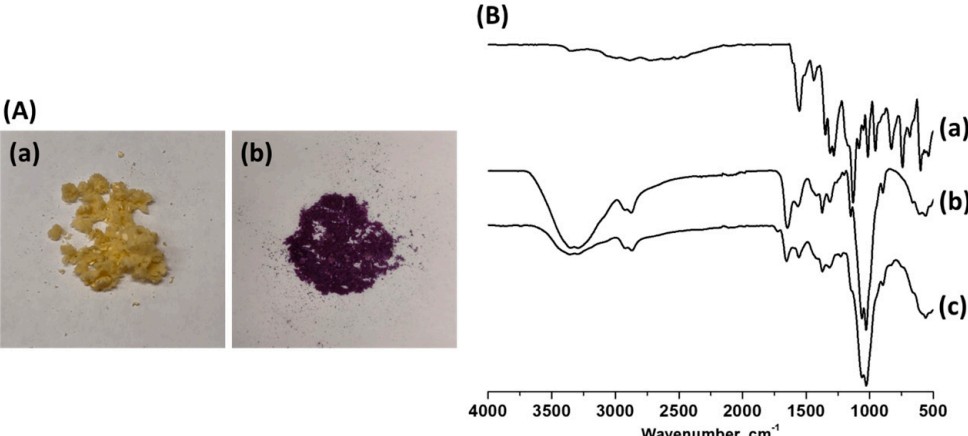

**Figure 14.** (**A**) Cross-linked chitosan (a) without dye and (b) with adsorbed dye, (**B**) FT-IR spectra of (a) pure phenol red dye, cross-linked chitosan (b) without dye and (c) with dye.

### 3.4. Mechanism for the Phenol Red Dye Reduction with Ag/Chitosan Catalyst

The possible mechanisms for the reduction of phenol red dye over the Ag/chitosan catalyst with NaBH$_4$ have been proposed and are shown in Figure 15. As shown in Figures 7B and 8, in the absence of the reducing agent, the phenol red dye gets adsorbed over the catalyst surface. Further, upon the addition of the reducing agent, the adsorbed dye got reduced and desorbed from the catalyst surface accordingly. It was also observed that NaBH$_4$ causes a reduction of the dye at a comparatively slow rate in the absence of catalysts or Ag metal (Figures 7A and 10A). Hence, Ag/chitosan catalysts facilitate not only the adsorption of dye but with Ag metal also induce the activation of the H$_2$ and reduction of adsorbed dye simultaneously. As shown in the proposed mechanism in Figure 15A, after the addition of both catalyst and NaBH$_4$, the dye gets adsorbed over the catalyst surface through surface-dye molecule interactions. The Ag metal further activates the H$_2$ originating from the decomposition of BH$_4^-$ anion, and the adsorbed dye is reduced accordingly, where the reduced dye desorbs from the catalyst surface instantly (Figure 14B). In this case, the Ag metal acts as a relay for the electron (or H$^-$), where it is relaying the electron from BH$_4^-$ anion to the acceptor dye molecule for subsequent reduction.

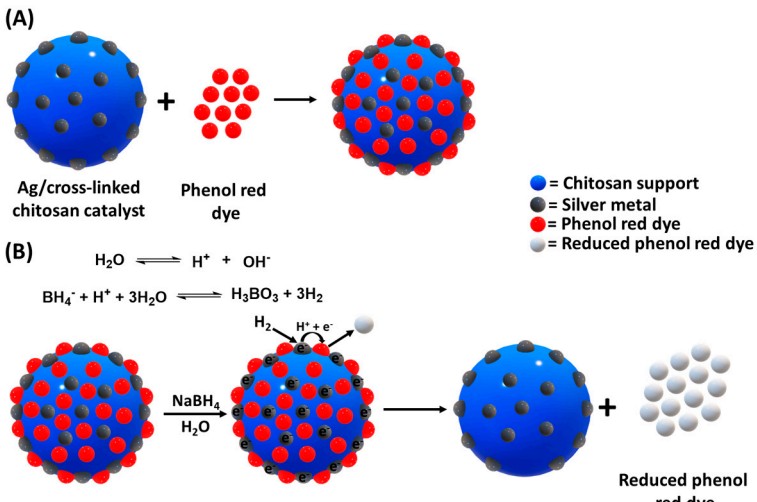

**Figure 15.** Possible mechanisms for the reduction of phenol red dye over Ag/chitosan catalyst (**A**) adsorption, and (**B**) reduction of phenol red dye with NaBH$_4$.

*3.5. Recyclability of Ag/Chitosan Catalyst upon Reduction of Phenol Red Dye*

The Ag/chitosan catalyst with 6.7 wt.%Ag metal loading was used for the recyclability study in the case of the phenol red dye reduction process. The recyclability of the catalyst was carried out five times, and the obtained recycled catalyst was further studied with ICP-OES analysis to quantify the loss of Ag metal during the recyclability study.

As shown in Figure 16, a minute decrease in the activity of the recycled catalyst was observed compared to the fresh catalyst during the recyclability study, whereas the catalyst showed identical activity during all the recycling steps. Further, the ICP-OES analysis study confirmed that 6.5 wt.% of Ag metal remained in the composition of the catalyst after the recyclability study. Hence, Ag/chitosan catalyst was found stable during the catalytic reduction of phenol red dye since no significant loss of Ag metal was observed.

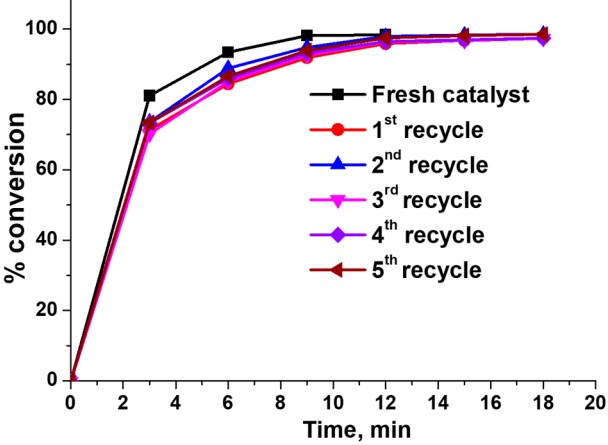

**Figure 16.** Recyclability study of the Ag/chitosan catalyst in catalytic reduction phenol red dye. Amount of 6.7 wt.%Ag/chitosan and NaBH$_4$ 6 and 5 mg, respectively.

## 4. Conclusions

The Ag/chitosan catalysts synthesised with cross-linked chitosan and Ag metal were studied in detail upon adsorption and catalytic reduction of phenol red dye in aqueous solutions. The Ag/chitosan catalyst prepared without any addition of the reducing agent showed excellent catalytic activity in phenol red dye reduction in the presence of NaBH$_4$. The activity of the catalyst was increased with increasing catalyst amount, NaBH$_4$, Ag metal loading and applied temperature. The *p*H of the reaction mixture in the range of

4–11 did not significantly influence the outcome as a complete reduction of the dye was observed and high catalytic activity. However, the dye reduction rate decreased excessively in a highly alkaline reaction mixture with *p*H 13.8, and no complete reduction of the dye was observed under such reaction conditions. The adsorption of the dye on the cross-linked chitosan support varied with the *p*H of the dye solution, where incomplete adsorption of dye was observed at *p*H 4 and 8 while a solution with nearly neutral *p*H, i.e., 6.4, facilitated efficient dye adsorption. No adsorption of the dye was observed in an alkaline dye solution with *p*H 11 and 13.8. The 6.7 wt.%Ag/chitosan catalyst was recycled five times in the dye reduction process, where identical catalyst activity was observed, and no significant loss of Ag metal was observed. Hence, in this report, a study regarding the adsorption as well as catalytic reduction of phenol red dye with cross-linked chitosan and Ag/chitosan catalyst, respectively, were studied where both processes were carried out at mild and economically feasible reaction conditions. Upon industrial-scale water purification, the concept might be feasible considering the low cost and easy availability of the Ag metal precursor as well as the use of renewable support material.

**Author Contributions:** C.C.S.: conceptualisation, visualisation, formal analysis, validation and writing-reviewing and editing; V.M.D.: resources, visualisation, formal analysis and validation; P.C.: supervision, writing-reviewing and editing, project administration and funding acquisition. J.-P.M.: supervision, writing-reviewing and editing, project administration and funding acquisition. S.G.K.: supervision, conceptualisation, investigation, resources, visualisation, validation, writing-original draft, project administration, writing-reviewing and editing. All authors have read and agreed to the published version of the manuscript.

**Funding:** This work is part of the activities by the Swedish Bio4Energy program and the Wallenberg Wood Science Center under the auspices of the Alice and Knut Wallenberg Foundation.

**Institutional Review Board Statement:** Not applicable.

**Informed Consent Statement:** Not applicable.

**Data Availability Statement:** No new data were created.

**Acknowledgments:** This work is part of activities of the Technical Chemistry, Department of Chemistry, Chemical-Biological Centre, Umeå University, Sweden. The Swedish Bio4Energy program and the Wallenberg Wood Science Center under the auspices of the Alice and Knut Wallenberg Foundation are gratefully acknowledged. This work is also a part of the activities of the Johan Gadolin Process Chemistry Centre at Åbo Akademi University.

**Conflicts of Interest:** The authors declare no conflict of interest.

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
