# Peer review of "Efficient Adsorption and Catalytic Reduction of Phenol Red Dye by Glutaraldehyde Cross-Linked Chitosan and Its Ag-Loaded Catalysts: Materials Synthesis, Characterization and Application"

_cleantechnol, doi:10.3390/cleantechnol5020024_

Round 1
Reviewer 1 Report
This paper reports Ag/chitosan as catalyst and adsorbent to remove phenol red dye from water. There is a lot of work in this paper, but I don't believe that it's worth publishing in current version. The followings are my serious concerns.
1. “The catalysts will be here onwards designated as 1.4 wt.% Ag/chitosan, 4.4 wt.% Ag/chitosan and 6.7 wt.% chitosan in upcoming descriptions regarding their characterisations and application in catalytic measurements.” In 2.3.2, the naming order of samples should correspond to the previous text, that is, the order of Ag content from large to small.
2. In 3.1, the analysis of FT-IR spectrum is inaccurate, the peak of N-H at 1500-1600 cm-1 should be bending vibration rather than stretching vibration.
3. Scheme 2 is wrong, Similarly, the scheme in Figure 14 is also wrong.
4. “As shown in Figure 6B, the rate of the reduction decreased with the amount of NaBH4 in the reaction mixture where about 60% of phenol red dye was reduced in 30 min when 1 mg of reducing agent has used.” This sentence in 3.2.2 is too complicated to understand.
5. In 3.2.2, why does phenol red dye still have such high discoloration rate without reductant? In this case, why was NaBH4 still included? This contradicts “NaBH4 possibly contributed mainly to the reduction process” in 3.2.4.
6. In 3.2.4, “the crosslinked chitosan showed lower reduction ability”. Is it the lower reduction ability (or it should be called catalytic capability) or the adsorption that causes the cross-linked chitosan to slowly discolor the solution?
7. When exploring the influence of initial pH of the phenol red dye solution, there was only one set of comparative experiments. Multiple groups of experiments with different pH should be set up.
8. Why is the peak at about 570 nm of the curve of 3 min, 6 min and 9 min in Figure 11A higher than the initial value, while the peak at about 570 nm of the curve of 3 min and 6 min adsorption in Figure 13B lower than the initial value?
9. Why is it that the disappearance of the characteristic bands of benzene ring, -OH, carbonyl (-C=O), C-O and sulfonate (SO3-) groups in phenol red dye molecules is due to the homogeneous distribution of dyes on the surface of cross-linked chitosan? Besides, how to prove that the dye is evenly distributed on the cross-linked chitosan surface?
10. The conclusion is too wordy and needs to be simplified.
11. There are some spelling mistakes in the article that need to be corrected.
Author Response
All the authors thank the reviewer for reviewing the manuscript. The response to the reviewer’s comments is given point by point and required changes have been also introduced in the manuscript.

Reviewer 2 Report
The submitted work is very interesting and useful piece of research and can be accepted for publication after minor corrections. Among them are the following.
1. The images and figures are a little bit blurry. The quality of figures should be improved.
2. What is the physical meaning of zones below zero on abscissa and ordinate in some figures?
3. Figures 11 and 16 with different symbols are hard to follow up as the values are very similar. Maybe different form of presentation of this data would be more appellative.
4. Similar can be stated about figures with adsorption and desorption as too many curves even with different colours are hard to follow up.
5. Did author tried to see the kinetics of reactions. Analysing the obtained results it looks as the first (or pseudo) first order reaction. It would be interesting to consider this aspect in their work too.
Author Response

(The authors gave the same response as above.)

Round 2
Reviewer 1 Report
It could be published in current version since the authors have addressed all the issues.
Author Response

(The authors gave the same response as above.)

Reviewer 2 Report
This work can be accepted for publication.